# First Brachial Cleft Anomalies in Children: An Innovative Surgical Technique Preventing External Auditory Canal Stenosis

**DOI:** 10.3390/children10071158

**Published:** 2023-07-01

**Authors:** Michal Kotowski, Jaroslaw Szydlowski

**Affiliations:** Department of Pediatric Otolaryngology, Poznan University of Medical Sciences, 60-572 Poznan, Poland

**Keywords:** branchial cleft, external auditory canal, stenosis, children

## Abstract

First branchial cleft anomalies (FBCAs) are rare congenital malformations that require complete surgical removal. A stenosis of the external auditory canal (EAC) may be the consequence of the disease and its treatment. The aim of this study is to present the details and results of an innovative surgical technique using part of the abnormality for reconstruction purposes. This study covered 28 surgically treated children with FCBA between 2014 and 2021. The analysis included the clinical manifestation form of the abnormality, presence of the EAC deformity, histopathological results, complications, and distant results. On the basis of Work’s classification system, 15 pediatric patients with type II FBCA and 13 children with type I FBCA were included in the further study. One child with type II FBCA and two with type I FBCA had a normal EAC. The preoperative appearance of the EAC was classified into three main types, each potentially accompanied by a skin ostium of the sinus/fistula in the EAC. Reconstruction with our technique was performed in 14 children (1 with type I FBCA and 13 with type II FBCA). Wound healing was uncomplicated in all cases. No recurrences were observed. This innovative surgical technique of the subtotal resection of FBCAs with simultaneous reconstruction is safe and prevents postoperative EAC stenosis. Despite the deliberate use of part of the abnormality wall for reconstructive purposes, it remained free of recurrences.

## 1. Introduction

A natural maturation of the branchial apparatus occurs between the fourth and eighth weeks of gestation [1]. Branchial cleft anomalies (BCAs) are rare congenital lesions resulting from disturbed developmental processes of branchial arches. BCAs are a heterogenous group that may have the form of a cyst, sinus, or fistula. Their location and course depend strictly on the type of lesion. Approximately 90% of them arise from the second cleft [2]. First branchial cleft anomalies (FBCAs) account for approximately 8% of all BCAs, being the second-most-common branchial cleft [3]. 

Potential clinical symptoms of FBCAs cover recurrent swelling of parotid or postauricular areas, abscess formation, otorrhea, deformation of the external auditory canal (EAC) (Figure 1), and persistent discharge from the cutaneous ostium located in retroauricular or preauricular region (buccal, submandibular). Owing to their heterogeneity, these abnormalities are commonly improperly diagnosed and treated [4,5]. It is estimated that at least 50% of FBCAs are initially misdiagnosed and treated with incisions and drainages instead of surgical excision [6].

There have been a few different classification systems discussed in the literature: Arnot’s, Work’s, and Olsen’s. Arnot was the first to classify these congenital lesions in 1971 [7]. That classification system included two types of FBCA. Type I was described as a painful cyst or discharging sinus in the parotid region, usually closely associated with the lower branches of the facial nerve. The type II abnormality included the cyst or sinus in the anterior triangle of the neck. The sinus could have the skin opening below the mandible, as well as a potential track extending to the EAC. On the contrary, Olsen’s proposal was based solely on the morphology of these abnormalities [8]. Hence, the author distinguished a form of a cyst, sinus, or fistula without any additional anatomical conditions. Probably the most popular classification is that proposed by Work in 1972 [9]. The author distinguished two types of FBCA based on anatomical and histological features. The first type of this lesion is typically localized in the retroauricular region, and it has a cystic form extending to the posterior or posteroinferior wall of the EAC. Histologically, type I FBCA consists solely of ectodermal components. A type II FBCA may be located anywhere from the EAC to the mandible angle or buccal area. Type II FBCAs are of ectodermal and mesodermal origin; therefore, some skin adnexal structures and cartilaginous elements are often found during histopathological examination.

Nevertheless, none of the commonly known classification systems have been found to be practically useful in surgical planning [10]. Despite many researchers’ efforts, the unpredictable relationship between the FBCA and the course of the facial nerve remains the major presurgical concern. Moreover, radiological imaging techniques have limited value for determining the conflict with neural structures. Another problem is the wide variety of clinical manifestations of FBCAs that is clearly visible in type II FBCAs. It is difficult to predict both anatomical interrelations and the extent of the resection. These make surgical treatment challenging. Due to these facts, a significant number of surgical decisions are made intraoperatively. 

Serious postoperative complications include the injury of the facial nerve (FN) and stenosis of the EAC [11,12,13,14]. The reported incidence of postsurgical FN paralysis differs between studies, ranging from 10% to 22% [11,13,14,15,16]. 

Recurrences have been observed in up to 22% of cases [13,14,17,18]. Complete surgical removal has been regarded as the treatment of choice in relation to the risk of recurrence [12,14,15].

EAC stenosis is a scarcely mentioned postsurgical complication in the literature, that stems directly from radical excision. How to combine a complete removal of the lesion with preservation of the adequate lumen of the EAC in particular cases is a dilemma for the surgeon. Facing this problem, we implemented the innovative technique of EAC reconstruction using part of the abnormality. This strategy based on histological knowledge sheds new light on the surgical treatment of FBCAs. The aim of this study is to present the details of the surgical technique and the distant results in our cohort of patients. An additional aim is to discuss the literature on the issue of postsurgical stenoses of EACs in children with FBCAs. 

## 2. Materials and Methods

Twenty-eight children with a first branchial cleft anomaly were surgically treated at the Department of Pediatric Otolaryngology between 2014 and 2021. A retrospective review of the clinical manifestation form of the abnormality, presence of the EAC deformity, intraoperative findings, histopathological results, complications, and post-op care, as well as distant results, was carried out. On the basis of the criteria of Work’s classification system, 15 pediatric patients with Work’s type II FBCA and 13 children with type I FBCA were included in the further study. 

Surgical technique: The aim of the technique was to provide an adequately wide and self-cleaning external auditory canal. The external U- (Figure 2), Y-, or S-shaped approach was implemented depending on the presence of a buccal or a mandibular mass and/or an opening of the sinus/fistula. After the dissection of the distal part of the abnormality (if present) or the opening of the duplicated EAC (Figure 3a), the relationship between the lumen of the EAC and the lumen of the abnormality was visualized (Figure 3b). Prior to the resection, a longitudinal or a spindle communicating incision along the party wall was carried out (Figure 4). Then, depending on the local situation, the defective skin of the EAC and the superior part of the abnormality (including cartilage if present) were resected (Figure 5). The superior part of the abnormality was removed adequately to the skin loss in the EAC. Generally, a resection of up to ½ of its circumference is required. The continuity of the EAC lumen was reconstructed (Figure 6) using the remaining part of the abnormality (usually its inferior wall) and stabilized with absorbable sutures remaining outside the lumen of the EAC (Figure 7). An ear gauze package with an ointment (*Oxytetracyclini hydrochloridum* and *Hydrocortisonum*) was placed in the external auditory canal for 3 weeks in each case. 

Subjective methods were implemented for the assessment of EAC diameter during follow-up. The postsurgical diameter of the EAC was compared with the contralateral one for each case separately. A set of ear speculums ranging from 2.5 mm to 5 mm was used. In addition, the otoendoscopy was performed in each case using a 4.0 mm rigid endoscope. A smaller EAC diameter compared with the contralateral side or less than 4 mm in general was considered as post-surgical stenosis. 

Statistical analysis was performed using MedCalc software (MedCalc Software Ltd., Version 20.118, Ostend, Belgium). The statistical significance level was determined with a *p* value of 0.05 or less. For qualitative indicators, the Chi-square test was used and presented as odds ratios. 

Moreover, a PubMed database search was performed to identify the articles concerning the issue of postsurgical EAC stenosis in FBCAs that were published between 2013 and 2023. The terms “first branchial cleft” and “stenosis” were used. The inclusion criteria regarding the search results were established as English language original research with the full text available. The exclusion criteria included the following: studies conducted on non-human participants, non-English language papers, papers that did not contain original research, case reports, reviews, meta-analyses, letters to editors, or those for which only an abstract was available. All articles that met the aforementioned criteria were screened for their applicability based on the full text content. 

## 3. Results

A left-sided anomaly was observed in 18 children and right-sided in 9 patients. A bilateral pathology was presented by one participant. 

The preoperative appearance of the external auditory canal was classified into three main types, each potentially accompanied by a skin ostium in the EAC. They presented as a plain, depressed, or elevated surface with or without EAC ostium of the sinus/fistula. Only 1 child out of 15 with type II FBCA and 2 individuals with type I FBCA had a normal EAC. The most common form of the visible EAC deformity was the elevation of the skin with the opening of the fistula/sinus, which was dominant among children with type II FBCA (47%). On the contrary, none of the patients with type I FBCA presented such a manifestation. The normal or depressed EAC with the skin ostium were the two most commonly observed in patients with type I FBCA. The other represented forms were depression and elevation, both without the ostium of the sinus or fistula (one and two participants, respectively). Two children manifested type II FCBAs in their EACs as a depression of the wall with a visible sinus or fistula ostium, and in another four, the EAC lumen was regular, but with a noticeable skin opening of the fistula or sinus. A summary of the clinical manifestation of FBCAs in the EACs is presented in Table 1. 

Owing to the risk of postsurgical EAC stenosis, the decision for reconstruction with the innovative technique (described above) was performed intraoperatively in 14 children. One of them represented type I FBCA (7.1% of all reconstructions), whereas thirteen of them had type II FBCA (92.9% of all reconstructions) (Table 2). A proportion of 86.7% of children with Work type II FBCA required EAC reconstruction. The statistical analysis confirmed the significant correlation between the presence of type II FBCA and the qualification to EAC reconstruction (*p* < 0.0001). 

Some interesting observations were made regarding the type of EAC deformation and the performed reconstructions. Children presenting with sinus or fistula ostium in the EAC required the reconstructive procedure significantly more often than those without such a communication (*p* = 0.0321). Similarly, patients with FBCA manifesting as the elevation of the inferior wall of the EAC were significantly more often operated upon using the described surgical reconstructive technique (*p* = 0.0469). The presence of the elevation in the EAC coexisting with the sinus or fistula ostium in the EAC was significantly correlated with the incidence of the reconstruction (*p* = 0.0027). On the contrary, the plain or depressed form of the EAC deformity was significantly less often connected to the necessity of reconstruction (*p* > 0.05).

The wound healing was uncomplicated in all cases. The follow-up period ranged from 12 to 48 months. It was longer than 24 months for 86% of children. The follow-up regimen included ENT examination with a meticulous comparison of the width of the lumen of both EACs in a patient every 3 months during the first year after the surgery, and every 6 months during the following 3 years. 

All the reconstructed EACs in children with FBCA were adequately wide compared with the contralateral ones. Any problems with extensive wax accumulation or suppuration were observed. No children presented recurrence of the disease during the follow-up. 

The initial search of the PubMed database revealed 293 articles. The final research material covered only five articles containing the results regarding EAC postsurgical stenosis. A summary, including the percentages of EAC stenoses, is presented in Table 3.

## 4. Discussion

FBCAs have neither a clinically useful classification system nor established guidelines. Despite being a demanding entity in surgical treatment, there is a lack of reports based on the representative study groups in the literature. The largest pediatric series was reported by Yang et al. in 2022 (109 pediatric and adult patients) [14], Chen et al. (100 children) [19], and Liu et al. in 2018 (70 children) [12]. Our group of patients consisted of a total of 28 children, and 13 of them were classified as Work’s type I and 15 as Work’s type II abnormalities. Yang et al., similarly to Liu et al., reported the domination of Work’s type I lesions (78% and 59% respectively) [12,14].

Yang et al. revealed puncta or masses in the EAC in 29.4% of patients [14]. Liu et al. highlighted the fact that each patient with FBCA presented an EAC abnormality [12]. In their observations, it may have presented as a promontory in the EAC, depression in the EAC, or a fistula. Our observations are different. The analysis revealed manifestations in the lumen of the EAC as promontory, depression, or normal (plain). Each of the three mentioned may coexist with an opening of the sinus or fistula in the EAC. The form of the EAC manifestation of the abnormality depended on the type of FBCA. 

As stressed previously in the literature, to provide complete resection in the region of the EAC, the abnormal skin or cartilage surrounding the lesion must be removed [12,13,14,19]. This is challenging in patients with the presence of a wide sinus or fistula ostium in the EAC or those with extremely tight adhesion of abnormal post-inflammatory tissues to the defective skin. Liu et al. warned against excessively extensive skin removal, as it may lead to granulation tissue formation and subsequent EAC stenosis [20]. All these authors proposed a simple suture of the skin in the EAC [12,13,14,19,20]. Nevertheless, in the case of extensive skin resection or uncontrolled rupture during dissection, it may be impossible to suture it without the narrowing of the EAC lumen. Li et al. reported that 47% of their patients had an EAC burst during the operation [13]. Unfortunately, there is a lack of data concerning the detailed surgical technique and sequelae in their cases. Chen et al. were required to suture the skin damage in 36% of cases [19]. Yang et al., Chen et al., and Li et al. reported no cases of EAC stenosis in their groups [13,14,19]. Liu et al. [12] revealed that almost 16% of their patients developed EAC stenosis after surgery, but surprisingly, it was in type I FBCAs, which is contrary to our observations [12]. This may have been the result of their special cartilage-splitting technique in type I FBCAs. 

It was identified in our study that the EAC manifestation is different in type I and type II FBCAs (Table 1). The results indicated that the presence of an elevation in the EAC coexisting with the sinus or fistula ostium in the EAC was significantly correlated with the fact of the reconstruction. Therefore, children with type I FBCAs rarely required reconstruction of the EAC (1 out of 13 patients). In our opinion, the cartilaginous elements in Work’s type II abnormality significantly interfere with the natural development and final appearance of the EAC. The radical excision including the inferior skin and/or cartilage wall of the EAC, especially in type II FBCAs, may result in postoperative canal stenosis due to skin loss and cicatrization. It may lead to disturbances in the EAC self-cleaning processes, keratitis obturans or external ear cholesteatoma formation. Therefore, we implemented the presented innovative surgical technique. This technique allowed for the elimination of this issue. The part of the anomaly used in reconstruction was lined with the squamous epithelium exposed to the lumen of the EAC. Despite deliberately using the part of the lesion for reconstruction purposes, no symptoms of recurrence have been observed. This stems from the fact that only an entrapped epithelium or adnexal structures in the surrounding tissues may create a problem of recurrence or suppuration. The technique not only prevents postoperative EAC stenosis, but also provides an excellent intraoperative view and supreme postsurgical EAC self-cleaning conditions. 

## 5. Conclusions

Our innovative surgical technique of subtotal resection of FBCAs with simultaneous reconstruction is safe and prevents postoperative EAC stenosis. Despite the deliberate use of part of the abnormality wall for reconstructive purposes, the patients remained free of recurrence. The presented technique is based on embryological and histological knowledge, and allows for a paradigm shift in our understanding of the extent of tissue resection in the surgical treatment of FBCAs.

## 6. Future Research Directions

The limited number of literature reports on the issue of EAC reconstruction in FBCA surgery indicates the necessity of further research. In our opinion, detailed prospective studies with the objective assessment of pre- and postsurgical volume measurements of the EAC are required. Another research direction should focus on the histopathological findings, and special attention should be paid to the type of epithelium of the abnormality and its changes during follow-up. The unsolved problem is the usefulness of the current classification systems in presurgical planning regarding the anatomical relationships of the FBCA and the facial nerve. We presume that the new classification system will stem from further research on FBCAs.

## Figures and Tables

**Figure 1 children-10-01158-f001:**
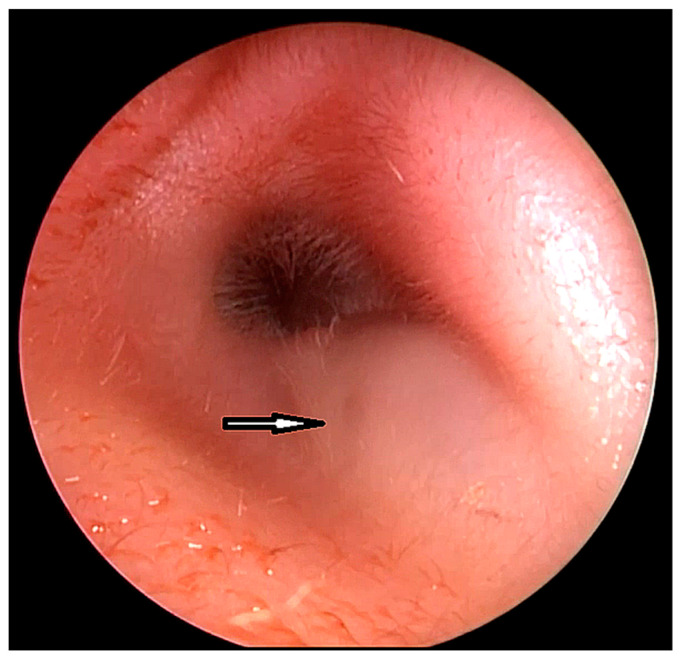
Preoperative endoscopic view of the external auditory canal: bulging of the skin with a visible sinus ostium (arrow).

**Figure 2 children-10-01158-f002:**
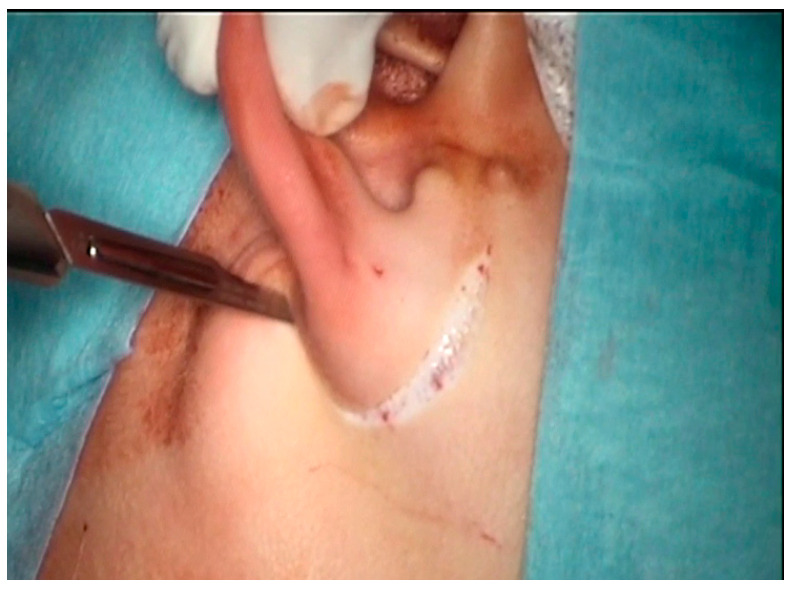
Skin incision.

**Figure 3 children-10-01158-f003:**
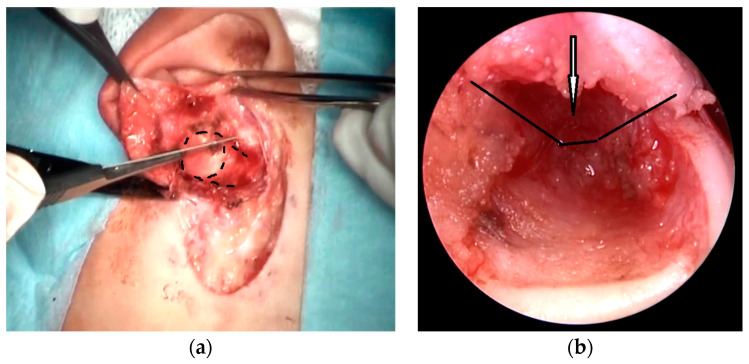
The anatomical relationship between the lumen of the EAC and the lumen of the abnormality: opening of the cartilaginous duplicated EAC (dotted line) (**a**); inside the duplicated EAC—endoscopic view on the party wall (arrow) (**b**).

**Figure 4 children-10-01158-f004:**
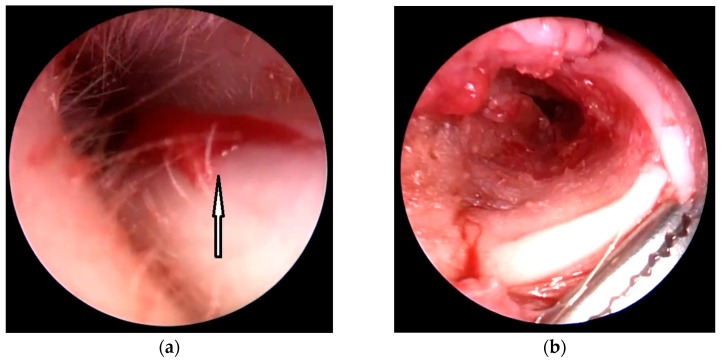
The longitudinal or spindle communicating incision along the party wall (arrows): endoscopic view from the EAC (**a**) and FBCA (**b**).

**Figure 5 children-10-01158-f005:**
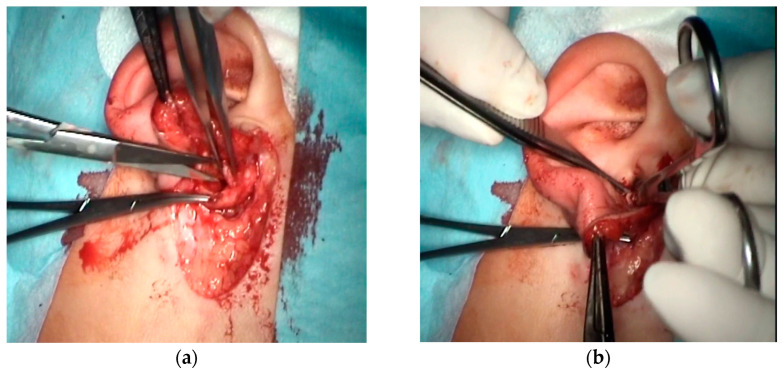
The resection of the defective skin +/− cartilage of the EAC (**a**) and the superior part of the abnormality (**b**).

**Figure 6 children-10-01158-f006:**
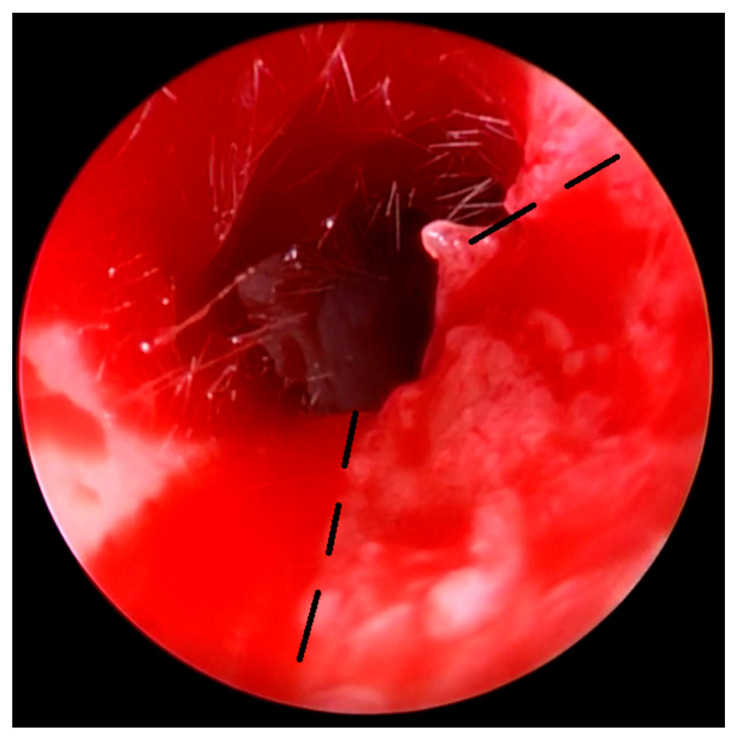
The lumen of the reconstructed EAC (endoscopic view)—the dotted lines indicate the margins of reconstructed area.

**Figure 7 children-10-01158-f007:**
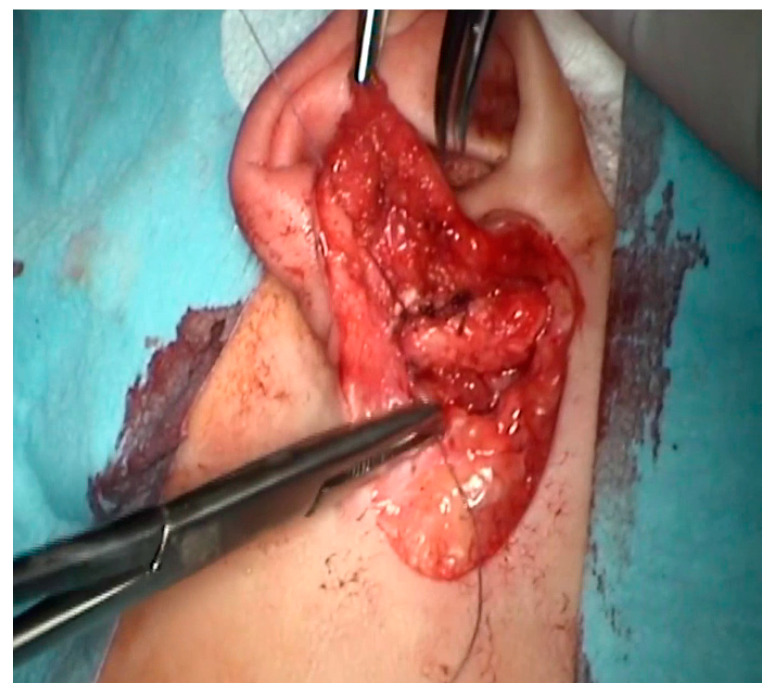
Reconstruction with stabilizing absorbable sutures.

**Table 1 children-10-01158-t001:** The type of the EAC deformation in FBCAs (Work types I and II).

Type of EAC Deformation	Type I FBCA	Type II FBCA
Plain	2	1
Plain with opening of fistula/sinus	4	4
Depression	1	1
Depression with opening of fistula/sinus	4	2
Elevation	2	0
Elevation with opening of fistula/sinus	0	7
Total	13	15

**Table 2 children-10-01158-t002:** The relationships among the type of the lesion, form of EAC deformation and performed EAC reconstruction.

	Sex	Side	Work Type	Type of Deformation	EAC Reconstruction
1	F	L	II	E *	Y
2	F	L	II	P *	Y
3	F	R	II	E *	Y
4	M	L	I	P *	N
5	F	L	I	P *	N
6	F	L	I	P	N
7	M	L	II	P *	N
8	F	L	I	D *	Y
9	M	R	II	E *	Y
10	F	R	II	P *	Y
11	F	R	II	P *	Y
12	F	R	II	E *	Y
13	F	L	II	E *	Y
14	F	R	II	D *	Y
15	M	L	II	E *	Y
16	M	L	I	E	N
17	F	R	I	P	N
18	F	L	I	P *	N
19	M	R	I	D	N
20	M	L	I	D *	N
21	F	L	I	P *	N
22	M	L&R	I	E	N
23	F	L	II	D	Y
24	F	L	I	D *	N
25	F	L	I	D *	N
26	F	R	II	E *	Y
27	F	L	II	P	N
28	M	L	II	D *	Y

Sex: M—male, F—female. Side: L—left, R—right. EAC reconstruction: Y—yes, N—no. Type of deformation: E—elevation, D—depression, P—plain, *—presence of fistula or sinus ostium in EAC.

**Table 3 children-10-01158-t003:** The overview of the publications presenting the results concerning postsurgical EAC stenosis (2013–2023, PubMed database search).

Authors [Reference]	Year of Publication	Number of Patients	EAC Stenosis (%)
Chen et al. [19]	2022	100	0
Yang et al. [14]	2021	109	0
Liu et al. [20]	2021	35	0
Liu et al. [12]	2018	70	15.7
Li et al. [13]	2017	30	0

## Data Availability

Data are available from the corresponding author on request.

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
