# Peer review of "First Brachial Cleft Anomalies in Children: An Innovative Surgical Technique Preventing External Auditory Canal Stenosis"

_children, 2023, doi:10.3390/children10071158_

Round 1

Reviewer 1 Report

I congratulate the authors for their work. and thank the opportunity to review the paper.

Words error remark:

1. line 55, 57 correct to FBCS

Additional remarks:

1. line 84- picture of endoscopy EAC is of low quality demostratinf blared or unfocused sinus

2. lines 127-128 why not use PRISMA guidlines for systemic literture review

3. short follow uo period - 15% of pt. are less than 1 year FU. if this can be improved q updated , i recommend it should.

Author Response

Responce to Reviewer #1

Reviewer’s comment: I congratulate the authors for their work. and thank the opportunity to review the paper.

Reply: I would like to thank the Reviewer for the revision.

Reviewer’s comment: Words error remark: 1. line 55, 57 correct to FBCS.

Reply: Thank You for this comment. The sentences have been corrected.

Reviewer’s comment: line 84- picture of endoscopy EAC is of low quality demostratinf blared or unfocused sinus

Reply: I improved the quality as much as possible. The sinus ostium is usually a punctum (visible on the dark spot on this picture which is a area of an extremely thin defective skin). In the majority of cases it is not a wide patent skin opening. The most important is a bulging of the inferior EAC wall visible in this figure. This manifestation is the easiest one to notice during otoscopy that raises the suspicion of FBCA. 

Reviewer’s comment: lines 127-128 why not use PRISMA guidlines for systemic literture review

Reply: I fully agree with your opinion that a systematic review would be of the greater value. Nevertheless, despite having some features of a systematic review this study does not meet all its criteria. This study was not designed as a systematic review, and it was not registered at inception. This original study focused on the problem of the preventive surgical technique of external auditory canal stenois in FBCAs. At this stage, I would prefer to keep the form of the narrative review in the methods and discussion, unless you find it unacceptable.

Reviewer’s comment: short follow uo period - 15% of pt. are less than 1 year FU. if this can be improved q updated , i recommend it should.

Reply: I think it is some kind of misunderstanding. It is clearly stated in the Results section: “The follow-up period ranged from 12 to 48 months. It was longer than 24 months in 86% of children.“. No patient with the FU period < 12 months was included to this study as we assumed 12 months was the minimal reasonable FU period.

Reviewer 2 Report

This is a study about an innovative surgical technique preventing external auditory canal stenosis in first brachial cleft anomalies. Fourteen children treated with the new procedure were included in the study.

The paper is well written. However, some issues remain.

The authors did not describe how EAC was reconstructed. Moreover, they must report differeneces with traditional procedure.

Comparison between traditional and new techniques must be added. Statistical analyses must be performed.

Author Response

Responce to Reviewer #2

Reviewer’s comment: This is a study about an innovative surgical technique preventing external auditory canal stenosisin first brachial cleft anomalies. Fourteen children treated with the new procedure were included in the study.

The paper is well written.

Reply: I would like to thank the Reviewer for the revision and for appreciating that the paper is well written.

Reviewer’s comment: However, some issues remain. The authors did not describe how EAC was reconstructed.

Reply: The EAC is reconstructed using the remaining part of the abnomality. This remaining part is adequate to the resected part of the defected skin and cartilage (if present) of the EAC. All the steps of the resection and reconstruction were described in the methods section and presented on the figures. Nevertheless, some more information has been added.

Reviewer’s comment: Moreover, they must report differeneces with traditional procedure. Comparison between traditional and new techniques must be added.

Reply: Thank you for this comment. Let me clarify the issue. The differences between traditional procedures and the proposed technique were presented in the Discussion section: As stressed previously in the literature, to provide complete resection in the region of the EAC, the abnormal skin or cartilage surrounding the lesion must be removed[12-14,19]. It is challenging in patients with the presence of wide sinus or fistula ostium in the EAC or these with extremely tight adhesion of abnormal post inflammatory tissues to the defective skin. Liu et al. warns against too extensive skin removal as it may lead to granulation tissue formation and the subsequent EAC stenosis [20]. All these authors propose a simple suture of the skin in the EAC [12-14, 19,20]. Nevertheless, in case of extensive skin resection or uncontrolled rupture during dissection, it may be impossible to suture it without narrowing of the EAC lumen. Li et al. reported that 47% of their patients had the burst of the EAC during the operation [13]. Unfortunately, there is a lack of data concerning detailed surgical technique and sequelae in their cases.” To summarize: all the authors proposed the total resection of the lesion despite consequences. Some of them sutured the remaining parts of the skin in the EAC (which may result in EAC stenosis and probably did but was not reported) but some leave the EAC as it was to the secondary healing. Up to date, no one proposed the EAC reconstruction using the part of the abnormality as the wall of the EAC instead of the resected skin or/and cartilage. The knowledge of the histology of FBCAs is the gist of the proposed technique.

Reviewer’s comment: Statistical analyses must be performed.

Reply: Statistical analyses have been added.

Round 2

Reviewer 1 Report

I accept the answer to my comments 

none

Author Response

Responce to Reviewer #1

Reviewer’s comment: I accept the answer to my comments

Reply: I would like to thank the Reviewer for the revision and acceptance.

Reviewer 2 Report

Thanks for implementing the paper. Two groups were compared (with and without reconstruction). However, the majority of the patients within each group had different malformations (type 1 vs type 2). Therefore, the groups are not comparable.

The authors stated that they analyzed the width of the EAC, but they did not included such results in the paper with adequate statistical analyses.

Author Response

Responce to Reviewer #2

Reviewer’s comment: Two groups were compared (with and without reconstruction). However, the majority of the patients within each group had different malformations (type 1 vs type 2). Therefore, the groups are not comparable.

Reply: With all due respect, I understand your doubts but I can not agree. First and foremost the paper presents the cohort of children with FBCA and describes the details of surgical technique and results in children requiring the EAC reconstruction preventing postsurgical stenosis. That was the aim of the study. Second, the study was not intended to compare the participants with and without reconstruction in any aspect. We proved that the type of the FBCA (according to Work’s classification system) was connected with the higher probability of EAC reconstruction. It may result from the different manifestations of the abnormality in the EAC depending on the type of FBCA. Nevertheless, both types may be connected with the significant EAC deformity and require reconstruction. I would like to emphasize that Woks’ classification system is based on histology. It can’t be applied to all the aspects of the surgery as EAC may be involved in both types. If you found it unacceptable, the only option is to remove one patints with type I FBCA whose EAC was reconstructed but it won’t change the statistical results.

Reviewer’s comment: The authors stated that they analyzed the width of the EAC, but they did not included such results in the paper with adequate statistical analyses.

Reply: The precise measurements of the EAC in a pediatric population or in individual patients has long been a challenge. The main issues remain the variability of the EAC location, shape, and curvature as well as the lack of non-radiological assessment tools. Addtionaly, there were the age differences in the study group and no indications for the routine post surgical radiological diagnostics. Thus, subjective methods were implemented. The postsurgical diameter of the EAC was compared to the contralateral one for each case separately. The set of ear speculums ranging from 2.5mm to 5mm was used. In addition, the otoendoscopy was performed in each case using 4.0mm rigid endoscope. The smaller EAC diameter comparing to contralateral side or less than 4 mm in general was considered as post surgical stenosis. No patient who underwent EAC reconstruction presented post surgical EAC stenosis. Therefore, the statistical analysis was not performed. I have added the description of the methodology regaring to the EAC diametr assessment to the paper. I would like to highlight that the aim of the study was not to compare the patients who underwent reconstruction to those who did not.

Round 3

Reviewer 2 Report

Please discuss that  the majority of the patients within each group had different malformations (type 1 vs type 2) in the Discussion section.

Author Response

Responce to Reviewer #2

Reviewer’s comment: Please discuss that  the majority of the patients within each group had different malformations (type 1 vs type 2) in the Discussion section.

Reply: Thank you for this suggestion. More content on this issue has been added in the Discussion section.